# COVID-19 Vaccination Is Associated with a Better Outcome in Acute Ischemic Stroke Patients: A Retrospective Observational Study

**DOI:** 10.3390/jcm11236878

**Published:** 2022-11-22

**Authors:** Pier Andrea Rizzo, Simone Bellavia, Irene Scala, Francesca Colò, Aldobrando Broccolini, Riccardo Antonica, Francesca Vitali, Benedetta Maria Angeloni, Valerio Brunetti, Riccardo Di Iorio, Mauro Monforte, Giacomo Della Marca, Paolo Calabresi, Marco Luigetti, Giovanni Frisullo

**Affiliations:** 1UOC Neurology, Catholic University of Sacred Heart, 00168 Rome, Italy; 2UOC Neurology, Fondazione Policlinico Universitario Agostino Gemelli, Istituto di Ricovero e Cura a Carattere Scientifico (IRCCS), 00168 Rome, Italy

**Keywords:** COVID-19, vaccination, SARS-CoV-2, ischemic stroke, outcome

## Abstract

Background: It is unclear whether and how COVID-19 vaccination may affect the outcome of patients with acute ischemic stroke (AIS). We investigated this potential association in a retrospective study by comparing previously vaccinated (VAX) versus unvaccinated (NoVAX) stroke patients. Methods: We collected clinical reports for all consecutive AIS patients admitted to our hospital and evaluated the outcome predictors in VAX and NoVAX groups. Adjustments were made for possible confounders in multivariable logistic regression analysis, and adjusted hazard ratios were calculated. Results: A total of 466 AIS patients (287 VAX and 179 NoVAX) were included in this study. The NIHSS score at discharge and mRS score at a 3-month follow-up visit were significantly lower in VAX patients compared to NoVAX patients (*p* < 0.001). Good outcomes (mRS 0–2) were significantly associated with COVID-19 vaccination before AIS (adjusted hazard ratio, 0.400 [95% CI = 0.216–0.741]). Conclusions: The observation that COVID-19 vaccination can influence the outcome of AIS provides support for further studies investigating the role of immunity in ischemic brain damage.

## 1. Introduction

Since the outbreak of the coronavirus disease 2019 (COVID-19) pandemic, the national healthcare system and economic markets have suffered a heavy backlash. The rapid growth of infected and deceased patients due to SARS-CoV-2 and the consequent efforts of pharmaceutical companies resulted in the rapid development of vaccines against SARS-CoV-2, such as Comirnaty (BNT162b2, Pfizer-BioNTech, Mainz, Germany), Spikevax mRNA-1273 (Moderna, Cambridge, MA, USA), Vaxzevria (ChAdOx1 nCov-19, AstraZeneca, Cambridge, UK) and a recombinant adenovirus type 26 vector encoding SARS-CoV-2 spike protein (Ad26.COV-2.S, Johnson & Johnson/Janssen, Beerse, Belgium), with more than 11 billion vaccines administered up until September 2022 [1]. On the other hand, the rapid development, authorization, distribution, and administration of COVID-19 vaccines aroused in some circles of public opinion, suspicions and doubts about their efficacy and safety. This was partly due to often unclear media communications and to the nature of the vaccine, considered a preventative measure that can lead to side effects and adverse reactions for healthy subjects. Although side effects such as vaccine-induced immune thrombotic thrombocytopenia (VITT), immune-mediated headache, cerebral venous sine thrombosis, anaphylactic shock, myocarditis, pericarditis, Guillain–Barrè syndrome and capillary leak syndrome have been described [2,3], phase-3 trials did not evidence an increase in cardiovascular or neurological events [4,5,6,7]. Some studies conducted in France [8], the US [9] and Israel [10] have analyzed the incidence of pulmonary embolism, myocardial infarction or cerebrovascular events without reporting a significant increase. In the UK, a study on the whole population found increased rates of intracranial venous thrombosis (ICVT) and thrombocytopenia in adults aged <70 years who received ChAdOx1-S vaccination compared to BNT162b2 [11]. Although the benefit for curbing the disease course and the side effects are clear, not enough research has investigated whether COVID-19 vaccines can have favorable effects on other pathological conditions, such as acute ischemic stroke (AIS). Therefore, the aim of this study was to evaluate the outcome for AIS patients in previously vaccinated versus unvaccinated stroke patients.

## 2. Materials and Methods

### 2.1. Study Design and Population

In this retrospective observational study, we included consecutive adult patients with a primary diagnosis of AIS who were admitted to our hospital between 1 January 2021 and 31 December 2021. We excluded patients with a diagnosis of intracranial hemorrhage or subdural hemorrhage, subarachnoid hemorrhage and patients with ongoing SARS-CoV-2 infection, and any patients aged <18 years. This study was approved by the local ethics committee (ID: 4895, Prot. No. 0016531/22-12/05/2022).

### 2.2. Clinical Evaluation

All patients underwent baseline neurological examination using the National Institutes of Health Stroke Scale (NIHSS) [12], laboratory tests, brain imaging and cardiological workout. A COVID-19 nasopharyngeal swab was performed at admission and every 5 days during hospitalization to exclude concomitant SARS-CoV-2 infection. Demographic data, pre-stroke medical history, cerebrovascular risk factors, previous SARS-CoV-2 infection, COVID-19 vaccination status, flu vaccinations within the previous 3 years, type of vaccine and vaccination dates were collected. Measures of outcome were the NIHSS score at discharge and the functional outcome at 3 months, which was assessed using the modified Rankin Scale (mRS) in an outpatient follow-up (cerebrovascular disease clinic) [13]. Details of the retrospective data collection are shown in the Appendix A.

### 2.3. Statistical Analysis

Continuous data were summarized using the mean and standard deviation (SD) and median and interquartile range (IQR); categorical data were summarized using counts and percentages. The distribution was studied using the Kolmogorov–Smirnov test. The Mann–Whitney *U*-test, with exact significance, was used for the non-normal distribution of data. An independent *t*-test was chosen for the analysis of continuous variables between two sets of normally distributed data. Dichotomous variables were compared using Fisher’s exact and chi-squared (χ^2^) tests. Ordinal variables were analyzed with the Wilcoxon signed-rank test, utilizing the exact test if necessary. All data obtained in this study were regularly registered. The study population was divided into two subgroups: COVID-19-vaccinated patients (VAX) and non-vaccinated patients (NoVAX). In the univariate analysis, all variables were compared between the subgroups VAX and NoVAX. Successively, to adjust the effect size for potential confounders, a multivariate analysis was performed. Variables used in the univariate analysis were entered into a multivariate logistic regression analysis to determine adjusted odds ratios. The multivariate model was built by selecting variables for their significance in the univariate comparison (e.g., age, COVID-19 vaccination status, hypertension, obesity, previous stroke, patent foramen ovale, NIHSS and hospitalization for COVID-19 at 3-month follow-up) and for their clinical relevance (e.g., treatments, coagulopathy and chronic obstructive pulmonary disease (COPD)). The level of significance was set at *p* < 0.05. All statistical analyses were performed using the Statistical Package for Social Sciences (SPSS^®^) software version 22 (SPSS, Inc., Chicago, IL, USA).

## 3. Results

From 1 January 2021 to 31 May 2022, 1324 patients discharged from the stroke unit and the emergency neurology department with a diagnosis of AIS were assessed for study eligibility. According to the exclusion criteria, 275 patients were excluded due to hemorrhagic pathogenesis of stroke, 489 due to appropriate coding of acute stroke, 35 due to acute phase COVID-19 and 59 lost due to lack of response to the follow-up questionnaire three months after acute ischemic event. Hence, 466 AIS patients were included in this study: 179 NoVAX and 287 VAX. A flow diagram showing the enrolment process is represented in Figure 1.

As expected, VAX patients were slightly older than NoVAX patients, with a similar male/female ratio. Hypertension and COPD were significantly more frequent in the VAX than in the NoVAX group, while coagulopathy was slightly more frequent in the NoVAX group. Most patients in the VAX group had been vaccinated with BNT162b2 (Pfizer), followed by ChAdOx1-S (AstraZeneca), Spikevax (Moderna) and Ad26.COV-2.S (Johnson & Johnson/Janssen) vaccine. Most of the patients (156, 80%) experienced the ischemic event after the second administration of the COVID-19 vaccination, with a mean timelapse of 101 days (SD = 68).

Thirty-four patients (15%) had a stroke after the first dose of the COVID-19 vaccine, in a mean time of 43 days. Five patients (3%) experienced an ischemic event after the third dose of the vaccine, at 14 days after administration. Stroke severity on admission, assessed by NIHSS, was similar between the two groups.

Despite being older and with more comorbidities, VAX patients had significantly lower NIHSS scores at discharge (*p* < 0.001) and mRS scores at 3-month follow-up evaluations (*p* < 0.001). After mRS dichotomization, the number of patients with poor prognosis (mRS 3–6) in the NoVAX group was higher than in the VAX group (*p* = 0.007). Finally, the death rate was lower in the VAX group compared to the NoVAX group (*p* < 0.001). During the 3-month follow-up period, 14 (7%) of VAX patients and 11 (9%) of NoVAX patients were infected with SARS-CoV-2. As expected, the majority of NoVAX patients (54%) and a minority (14%) of VAX patients required hospitalization for COVID-19. No patients in either group died from COVID-19-related causes. The demographics, risk factors and procedural data of the VAX and NoVAX groups are detailed in Table 1.

Among risk factors, hypertension was significantly more frequent in patients with poor prognoses than those without (*p* = 0.004). Patients who received a COVID-19 vaccination prior to the cerebral ischemic event had a better prognosis than unvaccinated stroke patients (*p* = 0.006). Finally, while the COVID-19 prevalence was similar, hospitalization for COVID-19 was significantly more prevalent in patients with a worse outcome than in functionally independent patients. The demographics, risk factors and procedural data of patients with functional independence (mRS 0–2) versus patients with a poor outcome (mRS 3–6) are detailed in Table 2.

In the multivariate analysis, a good outcome was significantly associated with vaccination before AIS (OR = 0.400; 95% CI = 0.216–0.741). On the contrary, a poor outcome (mRS 3–6) was associated with age > 85 years (OR = 2.374; 95% CI = 1.085–5.193), NIHSS score > 4 after discharge (OR = 7.524; 95% CI = 3.552–15.938) or previous stroke (OR = 2.451; 95% CI = 1.056–5.689). Detailed results of the multivariate analysis are displayed in Appendix A, and a forest plot is reported in Figure 2.

## 4. Discussion

In this observational study, we observed better short-term (NIHSS at discharge) and long-term outcomes (mRS at 3 months) for acute stroke patients vaccinated against COVID-19 before an ischemic event than a non-vaccinated stroke population. These data were even more significant considering the highest comorbidities, such as hypertension and COPD, in the VAX group. The old age of VAX patients is due to the COVID-19 vaccination campaigns, which prioritized the most fragile subjects, such as the elderly or patients with multiple comorbidities. In contrast, the mild prevalence of coagulopathies in NoVAX patients is justified by the initial exclusion of these patients from COVID-19 vaccination due to the increased risk of venous thrombosis or thromboembolic complications.

Considering the outcome predictors, we observed that an age above 85 years, NIHSS > 4 after discharge and previous stroke are significantly associated with a worse prognosis and a greater functional dependence of the patient. Conversely, the presence of an active vaccination against COVID-19 at the time of onset of an ischemic stroke constitutes the only predictor of a favorable prognosis and functional independence, regardless of SARS-CoV-2 infection or hospitalization for COVID-19 in the 3-month follow-up period.

While age and stroke severity are well-known outcome predictors [14], the same cannot be said for COVID-19 vaccination and, more generally, for vaccines.

COVID-19 RNA vaccines are able to induce balanced B- and T-cell responses, which are further increased by a second booster dose [15,16,17]. In addition to the expected stimulation of B cells, with a strong increase in B-cell percentage in peripheral blood and the production of specific antibodies, COVID-19 vaccination leads to the activation of T-cell-mediated immunity and to the differentiation of CD4+ T cells towards the Th1 response, which mediates proinflammatory functions aimed at the development of cell-mediated immune responses [18,19,20,21,22,23]. Limited information is available on the effect of the COVID-19 vaccine on regulatory lymphocytes, which play a critical role in immune homeostasis and immunological tolerance. However, although based on a few patients, it seems likely that CD4+ regulatory T (Treg) cells are increased following vaccination, performing an immunomodulatory function of the proinflammatory response [24].

Compelling evidence suggests a key role of the immune system, too, in the development, progression and, consequently, prognosis of ischemic stroke [25,26]. Brain ischemia triggers and, at the same time, influences from the local and systemic immune responses result in the induction of systemic immunosuppression, which increases the infectious risk [27], and an autoimmune response, which may further exacerbate brain injury [28]. In experimental models, ischemic stroke induces a Th1 polarization and promotes the production of proinflammatory cytokines, chemokines and reactive oxygen species and the disruption of the blood–brain barrier, inducing the evolution of ischemic brain injury [26]. On the contrary, high levels of Treg during the acute phase of ischemic stroke were independently associated with a good functional outcome at 3 months [29]. In our study, the administration of the SAR-CoV-2 vaccine and the consequent modification of the peripheral immune system could be the basis for a good outcome for VAX patients. On the basis of these observations, we could postulate a favorable effect on the prognosis of acute stroke patients with the COVID-19 vaccine thanks to its ability to induce a regulatory T response, which is able to suppress the damage related to the stroke proinflammatory response.

Another suggestion comes from the observation that in ischemic preconditioning, one or more short episodes of sublethal ischemia protect the brain against subsequent severe ischemic attacks [30]. These data were supported by the observation that ischemic preconditioning in a remote organ can lead to neuroprotection against brain ischemia due to a strong increase in circulating B cells, thereby reversing the reduction of the B-cell population after stroke [31].

As a result, the high percentage of circulating B cells observed in subjects undergoing COVID-19 vaccination may reveal the deleterious effects of the proinflammatory response caused by ischemic brain damage.

The main limitations of this study are the small sample size (at least partly due to the monocentric nature of this study), the impossibility of ruling out a selection bias and the duration of the follow-up having been limited to 3 months. Moreover, we did not evaluate previous SARS-CoV-2 infections before participants’ inclusion in the study.

## 5. Conclusions

Despite the small sample size and the short duration of the follow-up, the observation that the COVID-19 vaccination can influence the outcome of AIS provides support for further studies investigating the role of immunity in ischemic brain damage. In particular, it will be challenging to develop vaccines that can modulate the regulatory and B-mediated immune response to improve the prognosis for AIS patients.

## Figures and Tables

**Figure 1 jcm-11-06878-f001:**
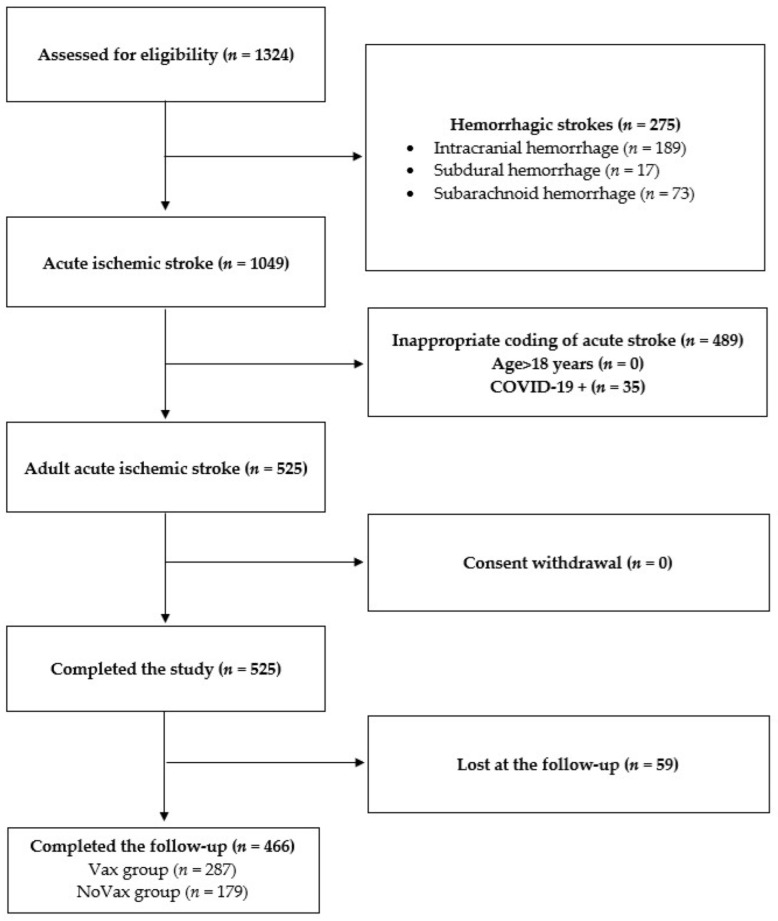
Participant screening flowchart. In this study, 1324 patients with acute stroke admitted to Fondazione Policlinico Universitario Agostino Gemelli were screened for participation eligibility. A total of 858 patients failed screening for the following reasons: (1) 275 patients were affected by hemorrhagic stroke; (2) 489 patients received a wrong or inappropriate coding of acute stroke; (3) no patients were <18 years; (4) 35 patients were COVID-19-positive. Finally, 59 patients were lost at the follow-up visit since they were untraceable. No patient declined to participate.

**Figure 2 jcm-11-06878-f002:**
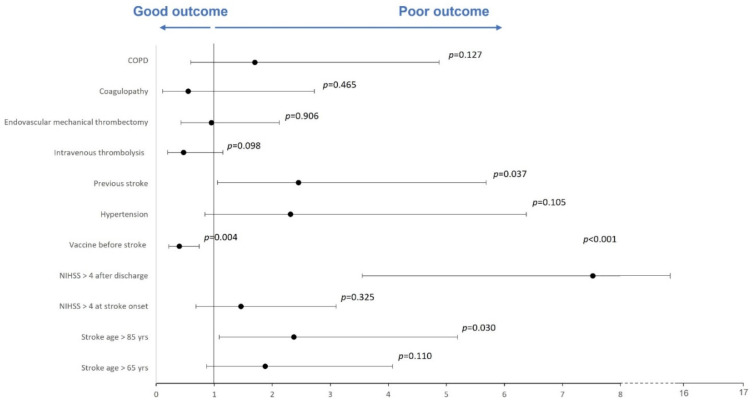
Forest plot of multivariate regression analysis evaluating outcome predictors. A forest plot of multivariable regression analysis evaluating risk factors, active COVID-19 vaccination, NIHSS score and age as predictors of a good or poor outcome. The Hosmer–Lemeshow test was not significant (*p* = 0.878). Nagelkerke’s R2 was 0.524.

**Table 1 jcm-11-06878-t001:** Demographic and clinical features of VAX and NoVAX groups.

Demographic Features and Risk Factors	VAX (*n* = 287)	NoVAX (*n* = 179)	*p*-Value
Age, median (years, IQR) ^§^	77 (14)	74.5 (22)	0.072
Sex, male, *n* (%) ^†^	161 (56)	89.5 (50)	0.311
Other vaccines (previous 3 years), *n* (%) ^†^	135 (47)	71 (40)	0.451
**Risk Factors**			
Hypertension, *n* (%) ^†^	**258 (90)**	**143 (80)**	**0.009**
Diabetes mellitus, *n* (%) ^†^	66 (23)	34 (19)	0.421
Hyperlipidemia, *n* (%) ^†^	109 (38)	64 (36)	0.698
Cigarette smoking, *n* (%) ^†^	20 (7)	21 (12)	0.102
Obesity, *n* (%) ^†^	11 (4)	7 (4)	1.000
COPD, *n* (%) ^‡^	**28 (10)**	**5 (3)**	**0.018**
Renal failure, *n* (%) ^†^	17 (6)	12 (7)	0.802
Atrial fibrillation, *n* (%) ^†^	54 (19)	30 (17)	0.577
Previous stroke, *n* (%) ^†^	34 (12)	21 (12)	0.891
Coagulopathy, *n* (%) ^‡^	**5 (2)**	**11 (6)**	**0.028**
Valvular heart disease, *n* (%) ^†^	20 (7)	13 (7)	0.952
Cardiopathy, *n* (%) ^†^	51 (18)	39 (22)	0.522
Patent foramen ovale, *n* (%) ^‡^	6 (2)	10 (6)	0.220
Cancer, *n* (%) ^†^	29 (10)	18 (10)	0.926
**Vaccine Type**			
BNT162b2, *n* (%)	207 (72)		
ChAdOx1-S, *n* (%)	43 (15)		
Ad26.COV-2.S, *n* (%)	12 (4)		
Spikevax, *n* (%)	25 (9)		
**Clinical Assessment and Procedures**			
Intravenous thrombolysis, *n* (%) ^†^	37 (13)	21 (12)	0.824
Endovascular mechanical thrombectomy, *n* (%) ^†^	48 (17)	29 (16)	0.765
NIHSS at stroke onset, median (IQR) ^§^	3 (7)	3 (7)	0.920
NIHSS after discharge, median (IQR) ^§^	**1 (4)**	**2 (6)**	**<0.001**
mRS after 3 months, median (IQR) ^§^	**1 (2)**	**2 (4)**	**<0.001**
mRS 3–6, *n* (%) ^†^	**97 (34)**	**87 (49)**	**0.006**
Death, *n* (%) ^†^	**20 (7)**	**43 (24)**	**<0.001**
COVID-19 during 3-month follow-up, *n* (%) ^†^	20 (7)	16 (9)	0.340
Hospitalization for COVID-19 during follow-up, *n* (%) ^†^	**3 (1)**	**9 (5)**	**0.029**

^†^: Pearson’s chi-squared test. ^‡^: Fisher’s exact test. ^§^: Mann–Whitney *U* test. COPD: Chronic obstructive pulmonary disease; NIHSS: National Institutes of Health Stroke Scale; mRS: modified Rankin Scale. Values which resulted statistically significant in the univariate analysis are highlighted in bold character.

**Table 2 jcm-11-06878-t002:** Demographic and clinical features of stroke patients according to good (mRS 0–2) and poor (mRS 3–6) outcomes.

Demographic Features and Risk Factors	mRS 0–2 (*n* = 289)	mRS 3–6 (*n* = 177)	*p*-Value
Age, median (years, IQR) ^§^	**74 (22)**	**79 (13)**	**<0.001**
Stroke age >65 years, *n* (%) ^†^	**196 (68)**	**155 (88)**	**<0.001**
Stroke age >85 years, *n* (%) ^†^	**49 (17)**	**54 (31)**	**<0.001**
Sex, male, *n* (%) ^†^	159 (55)	88 (50)	0.452
Vaccine before stroke, *n* (%) ^†^	**191 (66)**	**92 (52)**	**0.007**
Other vaccines (previous 3 years), *n* (%) ^†^	122 (42)	84 (47)	0.624
**Risk Factors**			
Hypertension, *n* (%) ^†^	**235 (81)**	**166 (94)**	**0.005**
Diabetes mellitus, *n* (%) ^†^	58 (20)	42 (24)	0.562
Hyperlipidemia, *n* (%) ^†^	101 (34)	71 (40)	0.078
Cigarette smoking, *n* (%) ^†^	29 (10)	12 (7)	0.314
Obesity, *n* (%) ^‡^	**6 (2)**	**12 (7)**	**0.026**
COPD, *n* (%) ^†^	17 (6)	16 (10)	0.133
Renal failure, *n* (%) ^†^	14 (5)	15 (9)	0.259
Atrial fibrillation, *n* (%) ^†^	43 (15)	41 (23)	0.057
Previous stroke, *n* (%) ^‡^	**14 (5)**	**41 (23)**	**0.037**
Coagulopathy, *n* (%) ^‡^	11 (4)	5 (3)	0.522
Valvular heart disease, *n* (%) ^†^	17 (6)	16 (9)	0.683
Cardiopathy, *n* (%) ^†^	55 (19)	35 (20)	0.825
Patent foramen ovale, *n* (%) ^‡^	**12 (6)**	**4 (2)**	**0.004**
Cancer, *n* (%) ^‡^	22 (8)	25 (14)	0.091
**Clinical Assessment and Procedures**			
Intravenous thrombolysis, *n* (%) ^†^	38 (13)	21 (12)	0.756
Endovascular mechanical thrombectomy, *n* (%) ^†^	40 (14)	37 (21)	0.098
NIHSS^2^ at stroke onset, median (IQR) ^§^	**2 (5)**	**6 (11)**	**<0.001**
NIHSS after discharge, median (IQR) ^§^	**1 (2)**	**5 (10)**	**<0.001**
NIHSS > 4 at stroke onset, *n* (%) ^†^	**95 (33)**	**120 (68)**	**<0.001**
NIHSS > 4 after discharge, *n* (%) ^†^	**37 (13)**	**106 (60)**	**<0.001**
COVID-19 during 3-month follow-up, *n* (%) ^†^	23 (8)	9 (7)	0.815
Hospitalization for COVID-19 during follow-up, *n* (%) ^‡^	**3 (1)**	**9 (5)**	**0.039**

^†^: Pearson’s chi-squared test. ^‡^: Fisher’s exact test. ^§^: Mann–Whitney *U* test. COPD: Chronic obstructive pulmonary disease; NIHSS: National Institutes of Health Stroke Scale; mRS: Modified Rankin Scale. Values which resulted statistically significant in the univariate analysis are highlighted in bold character.

## Data Availability

The corresponding author had full access to all the data in the study and takes responsibility for the integrity of the data and the accuracy of the data analysis.

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
