# Peer review of "COVID-19 Vaccination Is Associated with a Better Outcome in Acute Ischemic Stroke Patients: A Retrospective Observational Study"

_jcm, 2022, doi:10.3390/jcm11236878_

Round 1

Reviewer 1 Report

1.) Selection of patients: Why were intracerebral hemorrhage patients excluded? Why was "inappropriate coding as acute stroke" so frequent (n=489)? Are the results different if those patients are included (intention-to-treat anylysis)?

2.) Discussion is too much focussed on the immunological aspects which have not been studies here.

Author Response

Reviewer 1.

We thank the reviewer for all the suggestions.

  • Selection of patients: Why were intracerebral hemorrhage patients excluded? Why was "inappropriate coding as acute stroke" so frequent (n=489)? Are the results different if those patients are included (intention-to-treat analysis)?

we decided to include only patients with ischemic stroke because, in our sub intensive stroke unit, we essentially hospitalize ischemic strokes while, intracranial haemorrhages are hospitalized in other departments, whose clinical information is less complete. Furthermore, since the prognosis of patients with intracerebral haemorrhage is different from patients with ischemic stroke, the inclusion of this subgroup would have created a bias, or, would have required a significant increase in sample size.

The "inappropriate coding as acute stroke" is due to inappropriate ICD-9 coding at discharge. For this reason, patients with chronic vascular leukoencephalopathy, carotid or vertebral stenosis, small vessel disease, etc. have been wrongly coded as acute strokes. Not being patients with cerebral ischemia, they have not been included in the stroke clinical path, and, consequently, we do not have 3-month follow-up data available.

  • Discussion is too much focussed on the immunological aspects which have not been studies here.

We focused the discussion on strictly immunological aspects in order to try to justify the possible effect of the vaccine on the outcome of stroke. In any case, we have added other more strictly clinical or epidemiological aspects into discussion, as follows:

“The old age of VAX patients is obviously due to the COVID-19 vaccination campaigns, which gave priority to the most fragile subjects, such as the elderly or patients with multiple comorbidities. In contrast, the mild prevalence of coagulopathies in NoVAX patients is justified by the initial exclusion of these patients from COVID-19 vaccination, due to the increased risk of venous thrombosis or thromboembolic complications.”

Reviewer 2 Report

This paper addresses the question of COVID-19 vaccination is a predictor of good outcome in acute ischemic stroke (AIS): a retrospective observational study. The study included 466 patients with AIS, including 287 vaccinated and 179 unvaccinated. In conclusion, the authors concluded that COVID-19 vaccination can influence the outcome of AIS can provide support for further studies investigating the role of immunity in ischemic brain damage. The work brings new knowledge on the effect of vaccination on the course of cerebrovascular diseases and is carried out on a rather large group of patients which allows us to consider the statistical results reliable.

I would add one aspect before accepting for publication. In the discussion, it would be necessary to write a few sentences about the possible possibility of vascular incidents (and headaches that may suggest them) after vaccination based on the publication: https://pubmed.ncbi.nlm.nih.gov/35361131/

Author Response

Reviewer 2.

We thank the reviewer for all the suggestions.

  • I would add one aspect before accepting for publication. In the discussion, it would be necessary to write a few sentences about the possible possibility of vascular incidents (and headaches that may suggest them) after vaccination based on the publication: https://pubmed.ncbi.nlm.nih.gov/35361131/

We modified the text as follows:

“Although side effects as vaccine-induced immune thrombotic thrombocytopenia (VITT), immune-mediated headache, cerebral venous sine thrombosis, anaphylactic shock, myocarditis, pericarditis…”

Castaldo M, Waliszewska-ProsóÅ‚ M, Koutsokera M, et al. Headache onset after vaccination against SARS-CoV-2: a systematic literature review and meta-analysis. J Headache Pain. 2022;23(1):41. Published 2022 Mar 31. doi:10.1186/s10194-022-01400

Round 2

Reviewer 1 Report

Selection bias is not excluded by your study strategy. This should be mentionned in Limitations.

Author Response

We thank the reviewer for the suggestions.

Selection bias is not excluded by your study strategy. This should be mentionned in Limitations.

We have added this limitation in the “Conclusions” section of the paper, as follows: “The main limitations of this study are the small sample size, at least partly due to the monocentric type of this study, the impossibility of ruling out a selection bias and the duration of the follow-up limited to 3 months”
